# Unusual Tetrahydropyridoindole-Containing Tetrapeptides with Human Nicotinic Acetylcholine Receptors Targeting Activity Discovered from Antarctica-Derived Psychrophilic *Pseudogymnoascus* sp. HDN17-933

**DOI:** 10.3390/md20100593

**Published:** 2022-09-22

**Authors:** Xuewen Hou, Changlong Li, Runfang Zhang, Yinping Li, Huadong Li, Yundong Zhang, Han-Shen Tae, Rilei Yu, Qian Che, Tianjiao Zhu, Dehai Li, Guojian Zhang

**Affiliations:** 1Key Laboratory of Marine Drugs, Chinese Ministry of Education, School of Medicine and Pharmacy, Ocean University of China, Qingdao 266003, China; 2Illawarra Health and Medical Research Institute (IHMRI), University of Wollongong, Wollongong, NSW 2522, Australia; 3Laboratory for Marine Drugs and Bioproducts, Pilot National Laboratory for Marine Science and Technology, Qingdao 266237, China; 4Open Studio for Druggability Research of Marine Natural Products, Pilot National Laboratory for Marine Science and Technology, Qingdao 266237, China; 5Marine Biomedical Research Institute of Qingdao, Qingdao 266101, China

**Keywords:** psychrophilic fungus, tetrapeptides, tetrahydropyridoindoles unit, solid-phase synthesis, nicotinic acetylcholine receptors, molecular docking

## Abstract

Chemical investigation of the psychrophilic fungus *Pseudogymnoascus* sp. HDN17-933 derived from Antarctica led to the discovery of six new tetrapeptides psegymamides A–F (**1**–**6**), whose planar structures were elucidated by extensive NMR and MS spectrometric analyses. Structurally, psegymamides D–F (**4**–**6**) possess unique backbones bearing a tetrahydropyridoindoles unit, which make them the first examples discovered in naturally occurring peptides. The absolute configurations of structures were unambiguously determined using solid-phase total synthesis assisted by Marfey’s method, and all compounds were evaluated for their inhibition of human (h) nicotinic acetylcholine receptor subtypes. Compound **2** showed significant inhibitory activity. A preliminary structure–activity relationship investigation revealed that the tryptophan residue and the C-terminal with methoxy group were important to the inhibitory activity. Further, the high binding affinity of compound **2** to h*α*4*β*2 was explained by molecular docking studies.

## 1. Introduction

Psychrophilic fungi are a group of cold-adapted fungi residing in polar regions, alpine permafrost, glaciers, deep oceans, and other habitats [1,2,3,4,5,6], which are known for long-term low temperature, strong ultraviolet radiation, low nutrient and water availability, and frequent freeze and thaw cycles [7,8,9]. In order to adapt to such harsh conditions, these fungi have evolved special strategies in their metabolism and physiology [7,10,11,12], which endows the ability to produce diversified secondary metabolites, and makes psychrophilic fungi competitive microorganism group to serve structurally novel and bioactive natural products for drug development [13,14,15,16].

So far, a large number of novel natural products have been isolated from psychrophilic fungi, such as trisorbicillone A, a novel sorbicillin trimer that showed cytotoxic against HL60 cell lines (IC_50_ 3.14 μM) from deep-sea fungal strain *Phialocephala* sp. [17]; brevione A, the first breviane spiroterpenoid family from *Penicillium brevicompactum* [18]; penilactones A and B, novel polyketides with NF-kB inhibitory activity (inhibitory rate of 40% at 10 Mm) from *Penicillium crustosum* PRB-2 [19]; 16-membered trichobotryside A inhibiting the larvae settlement of Balamus amphitrite (EC_50_ 2.5 μg/mL) and 18-membered trychobotrysides B-C from *Trichobotrys effuse* [20]; and cytotoxic diterpenes conidiogenones B–G isolated from *Penicillium* sp, and conidiogenone C displayed significant cytotoxicity against HL60 and BEL-7047 cell lines with IC_50_ values of 0.038 mM and 0.9 mM respectively [21].

During our ongoing research on searching bioactive structures from Antarctic-derived fungi, a psychrophilic strain *Pseudogymnoascus* sp. HDN17-933, isolated from sand samples collected from Fildes Peninsula, was chosen for study based on its unique HPLC-UV profile. Detailed chemical investigation on its fermentation products afforded six tetrapeptides psegynamides A–F with human nicotinic acetylcholine receptors (nAChRs) targeting activity. Among those structures, psegynamides D–F (**4**–**6**) represented the first group of naturally occurring tetrapeptides carrying tetrahydropyridoindole units in the backbones. Herein, we will describe the isolation, structural elucidation, solid-phase total synthesis, and biological activity evaluation of these new compounds.

## 2. Results and Discussion

The psychrophilic fungal strain *Pseudogymnoascus* sp. HDN17-933 was isolated from Antarctica. They can grow in an environment of −5 °C, and the optimum growth temperature is 15–18 °C. The studies of their secondary metabolites are infrequent, and only 44 secondary metabolites have been reported until the time of writing. In this paper, the strain was cultured under static conditions at 15 °C for 30 days in a rice medium. The EtOAc extract of strain *Pseudogymnoascus* sp. HDN17-933 (Appendix A) was fractionated and purified by vacuum chromatography on silica gel, octadecyl-silica (ODS), Sephadex LH-20, and HPLC to afford six tetrapeptides psegynamides A–F (**1**–**6**) (Figure 1).

Compound **1** named psegynamide A was obtained as a white amorphous powder. Its molecular formula was determined to be C_30_H_39_N_5_O_5_ based on the molecular ion peak at *m/z* 550.3025 [M + H]^+^ (calcd 550.3024) in the HRESIMS analysis, indicating 14 degrees of hydrogen deficiency. In the ^1^H-NMR spectrum of **1** (Table 1), the characteristic signals of NH were displayed at low-field *δ*_H_ 8.68 (1H, d, *J* = 9.5 Hz), 8.37 (1H, d, *J* = 7.8 Hz), and 8.13 (1H, d, *J* = 9.1 Hz). The signals of *α*-proton, the characteristic for amino acid residues, were exhibited at *δ*_H_ 4.56 (1H, m), 4.44 (1H, m), 4.30 (1H, m), and 4.18 (1H, m). Four methyl groups were displayed at *δ*_H_ 0.72 (3H, d, *J* = 6.8 Hz), 0.68 (3H, d, *J* = 6.8 Hz), 0.61 (3H, d, *J* = 6.8 Hz), and 0.48 (3H, d, *J* = 6.8 Hz). Analysis of the ^13^C-NMR and HSQC spectra indicated the presence of four carbonyl peaks assignable to amide carbonyl groups at *δ*_C_ 173.6, 171.1, 170.7, and 161.9. Four groups of *α*-proton signals were showed at *δ*_C_ 57.3, 57.1, 53.9, and 52.9. The appearance of these spectra was typical of peptides. Combined analyses of ^1^H-^1^H COSY, HMBC, HSQC, NOESY, and TOCSY spectra further revealed the presence of 1 × Trp, 2 × Val, and 1 × Phe residues (Table 1), which were in accordance with the result of Marfey’s analysis (Appendix A). In the HMBC spectrum (Figure 2), NH of Val (1) has a correlation signal with C-1 on Trp; NH of Val (2) has a correlation signal with C-1 on Val (1); NH of Phe has a correlation signal with C-1 on Val (2). Meanwhile, in the NOESY spectrum, the cross-peaks of NH in Val (1) and H-2 in Trp, NH in Val (2) and H-2 in Val (1), NH in Phe and H-2 in Val (2) further confirmed the linkage of amino acid residues by HMBC spectrum. Thus, the planar structure of compound **1** was established as Trp^1^-Val^2^-Val^3^-Phe^4^.

The advanced Marfey’s acidolytic method was considered to determine the absolute configurations of compound **1** [22]. After FDAA derivatization and assisted by HPLC analysis, the presence of *L*-Val, *D*-Val, and *D*-Phe in compound **1** (Appendix A) was confirmed. However, due to the destruction of Trp residue during the strong acid environment, we failed to detect the Trp-FDAA derivative from the hydrolytic mixture. Therefore, we turned to using alkaline hydrolytic to determine the absolute configuration of Trp residue. With the alkaline hydrolytic condition of 5 M LiOH at 110 °C for 16 h was finally adopted, compound **1** was successfully hydrolyzed, and the absolute configuration of Trp was determined as *L*. As analyzed from the HPLC profile, both *L*-Val and *D*-Val derivatives were detected in the acidolytic mixture of compound **1**, which makes it another challenge to determine the accurate sequence of the tetrapeptide. Hence, two possible stereoisomers, **1a** (*L*-Trp^1^-*L*-Val^2^-*D*-Val^3^-*D*-Phe^4^) and **1b** (*L*-Trp^1^-*D*-Val^2^-*L*-Val^3^-*D*-Phe^4^), were prepared using solid-phase total synthesis (Figure 1A). By comparing the experimental NMR spectrum with the synthesized NMR data, the absolute configuration of compound **1** (Figure 1B, Appendix A) was determined to be *L*-Trp^1^-*D*-Val^2^-*L*-Val^3^-*D*-Phe^4^, which was inconsistent with the synthesized stereoisomer **1b**.

Compound **2** named psegynamide B was obtained as a white amorphous powder. Its molecular formula was determined to be C_31_H_41_N_5_O_5_ based on the molecular ion peak at *m/z* 564.3172 [M + H]^+^ (calcd 564.3180) in the HRESIMS analysis, which showed 14 Da molecular weight (MW) surplus to compound **1**. The ^1^H and ^13^C NMR of **2** were similar to those of **1** (Table 1), except for the presence of a methoxy group. The ^1^H, ^13^C NMR, and HSQC spectra showed a signal at *δ*_C_ 52.4/*δ*_H_ 3.56, which was assignable to a methoxy group. In addition, the methoxy site was established unambiguously by an HMBC experiment, in which a long-range correlation between CH_3_ (*δ*_H_ 3.56) and C-1 (*δ*_C_ 172.6) in the Phe unit was observed. Combined analyses of ^1^H-^1^H COSY, HMBC, HSQC, and TOCSY spectra assigned the linkage of amino acid residues as Trp^1^-Val^2^-Val^3^-Phe^4^-OCH_3_. Marfey’s acidolytic analysis confirmed the presence of *L*-Val, *D*-Val, and *D*-Phe in compound **2**, which was a similar case to compound **1**. Then, the solid-phase total synthesis and Marfey’s alkaline hydrolytic method were also used to determine the absolute configurations of compound **2**. Finally, the absolute configuration of **2** was assigned to be *L*-Trp^1^-*D*-Val^2^-*L*-Val^3^-*D*-Phe^4^-OCH_3_.

Compound **3** named psegynamide C was isolated as a white amorphous powder. The HRESIMS analysis of compound **3** gave a hydrogen adduct [M + H]^+^ at m/z 566.2977 (calcd 566.2973), corresponding to the molecular formula of C_30_H_39_N_5_O_6_, which showed one more oxygen than compound **1**. In the ^1^H NMR spectrum (Table 1), two aromatic H-atom signals appearing at *δ*_H_ 6.97 (2H, overlap, m) and 6.59 (2H, overlap, m) were attributable to a 1,4-substituted phenyl. In addition, a characteristic signal of OH was observed at low-field *δ*_H_ 9.18 (1H, s). Careful analysis of NMR of compound **3** revealed a similar structure as compound **1**, except for the presence of Tyr residue instead of Phe residue. Thus, the planar structure of compound **3** was established as Trp^1^-Val^2^-Val^3^-Tyr^4^. Based on Marfey’s acid/alkaline hydrolytic analysis, as well as solid-phase total synthesis, the absolute configuration of **3** was assigned to be *L*-Trp^1^- *D*-Val^2^-*L*-Val^3^-*D*-Tyr^4^.

Compound **4** was obtained as a white amorphous powder. Its molecular formula was determined to be C_31_H_39_N_5_O_5_ on the basis of the molecular ion peak at *m/z* 562.3032 [M + H]^+^ (calcd 562.3024) in the HRESIMS analysis, indicating 15 degrees of hydrogen deficiency. Compared the 1D-NMR data with compound **1**, compound **4** has an additional methylene (*δ*_C_ 40.8) and the CH-5 of Trp in compound **1** was changed as a quaternary carbon. In addition, one more degree of unsaturation indicated a cyclic structure for compound **4**. With the assistance of COSY correlations of H_2_-5′/NH-2′, H-2/NH-2′ and H-2/H_2_-3, and the HMBC correlations from H_2_-5′ to C-4/C-2, the tetrahydropyridoindole residue was deduced. Compound **4** represented the first example naturally occurring peptide with a tetrahydropyridoindole moiety. On the basis of Marfey’s acid hydrolytic analysis and solid-phase total synthesis, the absolute configuration of **4** was assigned to be as alternating LDLD chirality and named psegynamide D.

Psegymamides E-F (**5**–**6**) were obtained as white amorphous powders with the molecular formulas of C_32_H_41_N_5_O_5_ and C_32_H_41_N_5_O_6_ by HRESIMS, respectively. The 1D NMR spectra of **5** and **6** (Table 2) indicated a skeleton similar to psegynamide D (**4**). The difference between **4** and **5** was the replacement of the hydroxide group by a methoxy group (*δ*_C_ 52.3/*δ*_H_ 3.57) in the Phe unit, which was confirmed by HMBC correlation from OCH_3_ (*δ*_H_ 3.57) and C-1 (*δ*_C_ 172.7). The difference between **4** and **6** was the presence of Tyr-OCH_3_ residue instead of Phe residue. Thus, the planar structures of **5**–**6** were established. Similarly, the absolute configuration of **5**–**6** was assigned as alternating LDLD chirality by Marfey’s method and solid-phase total synthesis and finally named psegynamides E-F, respectively.

Nicotinic acetylcholine receptors (nAChRs) belong to the ligand-gated ion channel superfamily [23,24]. There are a large number of nAChR subunits, such as *α*1-10, *β*1-4, *γ*, *δ*, and *ε*, which may mediate analgesic effects and adverse reactions [25,26]. Targeting specific nAChRs subtypes may reduce adverse reactions while maintaining efficient analgesia and becoming a real new analgesic target. All new compounds (**1**–**6**) were evaluated for their activity at ACh-evoked currents mediated by human (h) *α*1*β*1*εδ*, *α*1*β*1*γδ*, *α*3*β*2, *α*3*β*4, *α*4*β*2, *α*7, and *α*9*α*10 nAChRs, among which compound **2** showed significant inhibitory activity at all subtypes (>70% inhibition) except *α7* (Figure 3, Appendix A). Interestingly, compound **2** selectively inhibited (>98% inhibition, n = 8–11) the *α*1*β*1*εδ* and *α*1*β*1*γδ* subtypes. For comparison, the α-conotoxin GI peptide from the marine cone snail *Conus geographus* venom antagonizes ACh-evoked currents mediated by h*α*1*β*1*εδ* with a half-maximal inhibitory concentration (IC_50_) of 20 nM [27]. According to the structural features of **1**–**6**, the preliminary structure–activity relationship (SAR) of inhibitory activities was tentatively discussed. Generally, compounds **1–3** exhibited stronger activities than compounds **4**–**6**, indicating that the presence of tetrahydropyridoindoles unit decreases inhibitory activities. Psegynamide B (**2**) showed stronger activities than psegynamide A (**1**), suggesting that C-terminal replacement with the methoxy group was beneficial to the activity.

To explain the different inhibitory activities and obtain further insight into the mechanism of nAChRs inhibition, molecular docking studies were carried out to explore the possible binding modes of compounds **1–6** and key interactions with nAChRs. The crystal structure of h*α*4*β*2 (PDB code: 5KXI) was used for further docking. The results in the docking study matched well with the inhibition activities (Appendix A), compounds with lower calculated docking scores are considered to have higher binding affinities with the target. Among them, compound **2** showed the highest binding affinity to h*α*4*β*2 with the most negative free binding energy (−4.2 kcal/mol). Analysis for optimized binding conformation of compound **2** displayed that the hydrogen at N-6 in Trp interacts with Ser A180 through a hydrogen bond with a distance of 2.2 Å, and the Trp formed H-pi stacked bonds with Gln B50 with a distance of 4.0 Å (Figure 4). While the Trp residue was replaced by the tetrahydropyridoindoles unit, the H-H and H-pi bond was lost. In addition, the NH in Val (2) of compound **2** interacts with Asp A49 through a hydrogen bond with a distance of 2.5 Å. It is interesting to find that the Arg B45 may be important for the activity because of that NH_2_ in Trp and the C=O in Phe-OCH_3_ of compound **2** simultaneously interact with Arg B45 through a hydrogen bond with distance of 2.2 Å and 3.3 Å, respectively. Furthermore, considering non-steroidal anti-inflammatory drugs and narcotics (opioids) are currently the most commonly used analgesic drugs, these drugs exhibit limitations in efficacy, unwanted side effects, and the problem of drug abuse [28,29]. The above findings provided that compound **2** might be helpful in developing different analgesic drugs by inhibiting nicotinic acetylcholine receptors.

## 3. Materials and Methods

### 3.1. General Experimental Procedures

NMR spectra were obtained on JEOLJN M-ECP 600 MHz spectrometers, of which TMS was an internal standard. The optical rotation of new compounds was calculated in MeOH on a JASCOP-1020 digital polarimeter. By using an LTQ Orbitrap XL (Thermo Fisher Scientific, Waltham, MA, USA) mass spectrometer, HRESIMS data were obtained. The spray voltage, capillary voltage, and tube lens were 4.0 kv, 16 v, and 35 v, respectively. The capillary temperature was 275 °C with a sheath gas flow rate of 10 arb. unit, and FT full mass spectra were acquired in the positive ionization mode at a resolution of 30,000 with 100–1500 Da mass range. The crude extract of *Pseudogymnoascus* sp. HDN17-933 were analyzed by reversed-phase HPLC (5 × 250 mm YMC C18 column, 5 µm) with a linear gradient of MeOH (A) and 0.1% aqueous TFA (B) from 5% to 100% A over 60 min at a flow rate of 1 mL/min. Column chromatography was carried out using the following chromatographic substrates: silica gel (300–400 mesh; Qingdao Marine Chemical Industrials, Qingdao, China), Sephadex LH-20 (GE Healthcare, Bio-Sinences Corp, Piscataway, NJ, USA). HPLC of the Waters company equipped with a 2998 PDA detector was performed on an ODS column (YMC-Pack ODS-A, 10 × 250 mm, 5 μm, 3 mL/min). UV spectra were carried out on Waters 2487 developed by Waters Corporation, Milford, USA. All Fmoc-amino acids were purchased from GL Biochem Ltd. (Shanghai, China). 2-chlorotrityl chloride resin at 1 mmol scale was purchased from Tianjin Nankai Hecheng S&T Co., Ltd (Tianjin, China).

### 3.2. Fungal Material and Fermentation

The fungal strain HDN17-933 was isolated from Fildes Peninsula, Antarctica, and identified as *Pseudogymnoascus* sp. based on internal transcribed spacer DNA sequencing. The sequence is available with the accession number MZ268166 at Genbank. It has been submitted to the Key Laboratory of Marine Drugs, working under the Ministry of Education of China, School of Medicine and Pharmacy, Ocean University of China. The fungus was cultured under the static condition at 15 °C in winter for 30 days in 1 L Erlenmeyer flasks, each containing rice (80 g) and naturally collected seawater (Huiquan Bay, Yellow Sea) (120 mL).

### 3.3. Extraction and Purification

All fermentation broth (40 L) was extracted with MeOH, filtered, concentrated, and partitioned between EtOAc and H_2_O. The EtOAc extract was evaporated under reduced pressure to give a crude gum (18.5 g). Moreover, the extract was subjected vacuum chromatography on silica gel (200–300 mesh) and eluted with stepped gradient elution via DCM:MeOH (100:1–1:1) to yield eight combined fractions (Fr.1 to Fr.8). Fr.6 was further separated using ODS (MeOH: H_2_O; 30:80–100:0) to obtain Fr.6-1 to Fr.6-6. Then Fr.6-3 and Fr.6-4 were further subjected to a Sephadex LH-20 column and eluted with MeOH to provide subfractions (from Fr.6-3-1 to Fr.6-3-4 and from Fr.6-4-1 to Fr.6-4-5). Fr.6-4-2 was purified by HPLC eluted with MeCN-H_2_O (38:62) to give compounds **1** (4.2 mg, *t_R_* = 11.5 min), **2** (3.4 mg, *t_R_* = 7.2 min), and **5** (3.5 mg, *t_R_* = 8.0 min), respectively. Fr.6-3-2 was purified by HPLC eluted with MeOH-H_2_O (65:35) to give compounds **3** (5.1 mg, *t_R_* = 10.2 min), **6** (4.3 mg, *t_R_* = 11.0 min), and **4** (3.2 mg, *t_R_* = 13.5 min), respectively.

### 3.4. Physical and Chemical Data

Psegynamide A (**1**): white amorphous powder; [*α*]^20^_D_ + 30.2 (*c* 0.2, MeOH); UV (DAD) λ_max_ 218 nm, 283 nm; ^1^H and ^13^C NMR (DMSO-*d*_6_), see Table 1; HREIMS *m/z* 550.3025 [M + H]^+^ (cacld for C_30_H_40_N_5_O_5_, 550.3024).

Psegynamide B (**2**): white amorphous powder; [*α*]^20^_D_ + 23.8 (*c* 0.3, MeOH); UV (DAD) λ_max_ 217 nm, 283 nm; ^1^H and ^13^C NMR (DMSO-*d*_6_), see Table 1; HREIMS *m/z* 564.3172 [M + H]^+^ (cacld for C_31_H_42_N_5_O_5_, 564.3180).

Psegynamide C (**3**): white amorphous powder; [*α*]^20^_D_ + 25.3 (*c* 0.3, MeOH); UV (DAD) λ_max_ 217 nm, 282 nm; ^1^H and ^13^C NMR (DMSO-*d*_6_), see Table 1; HREIMS *m/z* 566.2977 [M + H]^+^ (cacld for C_30_H_40_N_5_O_6_, 566.2973).

Psegynamide D (**4**): white amorphous powder; [*α*]^20^_D_ − 14.7 (*c* 0.3, MeOH); UV (DAD) λ_max_ 217 nm, 282 nm; ^1^H and ^13^C NMR (DMSO-*d*_6_), see Table 2; HREIMS *m/z* 562.3032 [M + H]^+^ (cacld for C_31_H_40_N_5_O_5_, 562.3024).

Psegynamide E (**5**): white amorphous powder; [*α*]^20^_D_ − 17.8 (*c* 0.2, MeOH); UV (DAD) λ_max_ 216 nm, 282 nm; ^1^H and ^13^C NMR (DMSO-*d*_6_), see Table 2; HREIMS *m/z* 576.3186 [M + H]^+^(cacld for C_32_H_42_N_5_O_5_, 576.3180).

Psegynamide F (**6**): white amorphous powder; [*α*]^20^_D_ − 20.3 (*c* 0.2, MeOH); UV (DAD) λ_max_ 216 nm, 283 nm; ^1^H and ^13^C NMR (DMSO-*d*_6_), see Table 2; HREIMS *m/z* 592.3133 [M + H]^+^ (cacld for C_32_H_42_N_5_O_6_, 592.3130).

### 3.5. Advanced Marfey’s Analysis of Acid Hydrolytic for Val, Phe, Tyr

Compounds **1**–**6** (1.0 mg each) were reacted with 6 M HCl (1.5 mL) at 110 °C for 12 h; the hydrolysates were concentrated to dryness. The hydrolysates and standard amino acids (150 μM) were then successively treated with water (300 μL), FDAA (10 mg/mL solution in acetone, 100 μL), acetone (300 μL), and NaHCO_3_ (1 M, 150 μL) at 45 °C water bath heating for 2.0 h. Then, the reaction was stopped with HCl (2 M, 75 μL) prior to HPLC analysis. Amino acid standards were similarly derivatized with FDAA. The resulting FDAA derivatives of compounds **1**–**6**, *L*- and *D*-Val, *L*- and *D*- Phe, *L*- and *D*- Tyr were analyzed by HPLC eluted with a linear gradient of MeOH (A) and 0.10% aqueous TFA (B) from 40% to 100% in an over 45 min with UV detection at 340 nm.

### 3.6. Modified Marfey’s Analysis of Alkaline Hydrolytic for Trp

Considering that Trp FDAA derivatives were not detected in the resulting FDAA derivatives of compounds **1**–**3**, it might result from that Trp amino acid was hydrolytic and destroyed in strong hydrochloric acid. Therefore, Marfey’s alkaline hydrolytic method was considered to determine the absolute configuration of Trp residue. Through continuous attempts, the alkaline hydrolytic condition of 5 M LiOH at 110 °C for 16 h was finally adopted, and the absolute configuration of Trp was determined as *L*.

### 3.7. Solid-Phase Total Synthesis of 1a, 1b

Compounds **1a**, **1b** were synthesized by 2-chlorotrityl chloride resin (loading 1 mmol/g) as described in Figure 1. Then, 4 equiv of Fmoc-*D*-Phe (2 mmol) was added to a suspension of 1 equiv of 2-chlorotrityl chloride resin (0.50 g, 0.5 mmol), 4 equiv of HCTU (2 mmol), 8 equiv of DIEA (4 mmol), and DCM/DMF (*v*:*v* = 1:1). After stirring for 2 h, the resin was filtered and washed with DCM/DMF (1:1) for three times. The reaction mixture was treated with piperidine:DMF (*v*:*v* = 1:4) for stirring 30 min to deprotect the Fmoc group. After filtering and washing, the resin was used as resin-*D*-Phe^4^-NH_2_ for the next coupling reaction.

A solution of HCTU (2 mmol) and DIEA (4 mmol) in DCM/DMF (1:1) was added to a mixture of resin-*D*-Phe^4^-NH_2_, Fmoc-*L*-Val/ Fmoc-*D*-Val (2 mmol). After the reaction mixture was stirred for 60 min, the resin was filtered and washed. To remove the Fmoc group, the same procedure was repeated as above. The resin was used as resin-*D*-Phe^4^-*L*-Val^3^-NH_2_/*D*-Phe^4^-*D*-Val^3^-NH_2_ for the subsequent next coupling reaction, respectively. A solution of HCTU (2 mmol) and DIEA (4 mmol) in DCM/DMF (1:1) was added to a mixture of resin-*D*-Phe^4^-*L*/*D*-Val^3^-NH_2_, Fmoc-*D*-Val/ Fmoc-*L*-Val (2 mmol). After the reaction mixture was stirred for 60 min, the resin was filtered and washed. To remove the Fmoc group, the same procedure was repeated as above. The resin was used as resin-*D*-Phe^4^-*L*-Val^3^-*D*-Val^2^-NH_2_/*D*-Phe^4^-*D*-Val^3^-*L*-Val^2^-NH_2_ for the next coupling reaction, respectively.

A solution of HCTU (2 mmol) and DIEA (4 mmol) in DCM/DMF (1:1) was added to a mixture of resin-*D*-Phe^4^-*L*/*D*-Val^3^-*D*/*L*-Val^2^-NH_2_, Fmoc-*L*-Trp (Boc)-OH (2 mmol). After the reaction mixture was stirred for 60 min, the resin was filtered and washed. To remove the Fmoc group, the same procedure was repeated as above. The resin was used as resin-*D*-Phe^4^-*L*-Val^3^-*D-*Val^2^-*L*-Trp^1^(Boc)-NH_2_/*D*-Phe^4^-*D*-Val^3^-*L*-Val^2^-*L*-Trp^1^(Boc)-NH_2_, respectively. Subsequent to TFA cleavage (TFA: TIPS: H_2_O, 90:5:5, 3 h), the reaction solution was treated with cold ether to obtain precipitation. The precipitate was furtherly purified by HPLC to give the compounds **1a** (*L*-Trp^1^-*L*-Val^2^-*D*-Val^3^-*D*-Phe^4^) and **1b** (*L*-Trp^1^-*D*-Val^2^-*L*-Val^3^-*D*-Phe^4^), respectively.

### 3.8. Xenopus Laevis Oocyte Preparation and Microinjection

All procedures were approved by the University of Wollongong Animal Ethics Committees (project number: AE2003). Female *Xenopus laevis* were sourced from Nasco (Fort Atkinson, WI, USA), and a maximum of four frogs were kept in a 15 L aquarium at 20–26 °C with 12 h light/dark cycle. Oocytes were obtained from five-year-old frogs anesthetized with 1.7 mg/mL ethyl 3-aminobenzoate methanesulfonate (pH 7.4 with NaHCO_3_). Stage V-VI oocytes (Dumont’s classification; 1200–1300 μm diameter) were defolliculated with 1.5 mg/mL collagenase Type II (Worthington Biochemical Corp., Lakewood, NJ, USA) at room temperature for 1–2 h in OR-2 solution containing (in mM): 82.5 NaCl, 2 KCl, 1 MgCl_2_ and 5 HEPES at pH 7.4.

The human muscle nAChR clones (*α*1, *β*1, *γ*, *δ* and *ε*) were purchased from Integrated DNA Technologies (Coralville, IA, USA), whereas the human *α*3, *α*9, *α*10, *β*2, and *β*4 clones were purchased from OriGene (Rockville, MD, USA), and all were subsequently inserted into the pT7TS vector. The human *α*4 and *α*7 clones were obtained from Prof. Jon Lindstrom (University of Pennsylvania, Philadelphia, PA, USA). Plasmid constructs of the human nAChR clones were linearized for in vitro mRNA synthesis using mMessage mMachine transcription kit (AMBION, Forster City, CA, USA).

Oocytes were injected with 5 ng cRNA for h*α*1*β*1*γδ*, h*α*1*β*1*εδ*, h*α*3*β*2, h*α*3*β*4 and h*α*4*β*2, 10 ng cRNA for h*α*7 nAChR, and 35 ng cRNA for h*α*9*α*10 nAChR (concentration confirmed spectrophotometrically and by gel electrophoresis). The muscle subunit cRNA ratio was 2:1:1:1 (*α*1:*β*1: *γ*/*ε*:*δ*), whereas the heteromeric *α* and *β* subunit cRNA ratio was 1:1, injected using glass pipettes pulled from glass capillaries (3-000-203 GX, Drummond Scientific Co., Broomall, PA, USA). Oocytes were incubated at 18 °C in sterile ND96 solution composed of (in mM): 96 NaCl, 2 KCl, 1 CaCl_2_, 1 MgCl_2_, and 5 HEPES at pH 7.4, supplemented with 5% fetal bovine serum, 50 mg/L gentamicin (GIBCO, Grand Island, NY, USA) and 10000 U/mL penicillin-streptomycin (GIBCO).

### 3.9. Oocyte Two-Electrode Voltage Clamp Recording and Data Analysis

Electrophysiological recordings were carried out 2–5 days post cRNA microinjection. Two-electrode voltage clamp recordings of *X. laevis* oocytes expressing human nAChRs were performed at room temperature (21–24 °C) using a GeneClamp 500B amplifier and pClamp9 software interface (Molecular Devices, Sunnyvale, CA, USA) at a holding potential −80 mV. Voltage-recording and current-injecting electrodes were pulled from GC150T-7.5 borosilicate glass (Harvard Apparatus, Holliston, MA) and filled with 3 M KCl, giving resistances of 0.3–1 MΩ. Due to the Ca^2+^ permeability of *α*9*α*10 nAChRs, 100 µM BAPTA-AM incubation was carried out before recording to prevent the activation of *X. laevis* oocyte endogenous Ca^2+^-activated chloride channels. Oocytes expressing h*α*9*α*10 nAChRs were perfused with ND115 solution containing (in mM): 115 NaCl, 2.5 KCl, 1.8 CaCl_2_, and 10 HEPES at pH 7.4, whereas oocytes expressing all other nAChR subtypes were perfused with ND96 solution using a continuous Legato 270 push/pull syringe pump perfusion system (KD Scientific, Holliston, MA, USA) at a rate of 2 mL/min in an OPC-1 perfusion chamber of <20 µL volume (Automate Scientific, Berkeley, CA, USA).

Initially, oocytes were briefly washed with ND115/ND96 solution, followed by 3 applications of ACh at a half-maximal excitatory ACh concentration (EC_50_) for the nAChR subtypes (3 µM for h*α*4*β*2, 5 µM for h*α*1*β*1*γδ* and h*α*1*β*1*εδ*, 6 µM for h*α*3*β*2 and h*α*9*α*10, 100 µM for h*α*7 and 300 µM for h*α*3*β*4) [27]. Washout with bath solution was performed for 3 min between ACh applications. Oocytes were incubated with compounds for 5 min with the perfusion system turned off, followed by co-application of ACh and compound with flowing bath solution. All compound solutions were prepared in ND115/ND96 + 0.1% bovine serum albumin (BSA). Incubation with 0.1% BSA was performed to ensure that the BSA and the pressure of the perfusion system had no effect on nAChRs. Peak current amplitudes before (ACh alone) and after (ACh + compound) compound incubation were measured using Clampfit version 10.7.0.3 software (Molecular Devices, Sunnyvale, CA, USA), where the ratio of ACh + compound-evoked current amplitude to ACh alone-evoked current amplitude was used to assess the activity of the compounds at the nAChRs. All electrophysiological data were pooled (n = 6–11) and represent means ± standard deviation (SD). Data analysis was performed using GraphPad Prism 5 (GraphPad Software, La Jolla, CA, USA).

### 3.10. Molecular Docking

Molecular docking studies were performed using MOE 2016. The crystal structure of human *α*4*β*2 nAChR (PDB identifier: 5kxi) was obtained from the Protein Data Bank (http://www.rcsb.org, accessed on 28 September 2016). Prior to docking, heteroatoms and water molecules in the protein were removed. In the meantime, compounds were minimized. The result of each ligand was furtherly analyzed by using PyMOL.

## 4. Conclusions and Discussion

In conclusion, the genus *Pseudogymnoascus* is a group of psychrophilic fungi that possess good potential in serving structurally unique structures but not widely investigated. In this study, chemical investigation of secondary metabolites from the psychrophilic fungus *Pseudogymnoascus* sp. HDN17-933 led to the discovery of six new tetrapeptides psegynamides A–F (**1**–**6**), among which psegynamides D–F (**4**–**6**) was the first naturally occurring peptide bearing a tetrahydropyridoindoles moiety. To the best of our knowledge, only a few synthetic peptides bearing the tetrahydropyridoindoles motif have been reported [30,31], for example, tetrahydropyridoindoles as cholecystokinin and gastrin antagonists in 1992 [30]. Solid synthesis techniques assisted by Marfey’s method were employed to work out the absolute configuration, especially in solving the challenges of determining the order of *L*-Val and *D*-Val isomers in the peptide sequences. Moreover, compound **2** was found to have bioactivity by inhibiting human nAChRs. Considering that non-steroidal anti-inflammatory drugs and narcotics (opioids) are currently the most commonly used analgesic drugs, these drugs exhibit limitations in efficacy, unwanted side effects, and the problem of drug abuse. To overcome these problems, the discovery of different molecular participants in the pain pathways could bring new opportunities for therapeutic intervention. Compound **2** might be helpful in developing potential natural short peptide inhibitor of nAChRs from Antarctica-derived fungus. The above findings illustrate and highlight the validity of exploiting psychrophilic fungus for the discovery of structurally novel and bioactive natural products.

## Data Availability

Not applicable.

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
