# Peer review of "Unusual Tetrahydropyridoindole-Containing Tetrapeptides with Human Nicotinic Acetylcholine Receptors Targeting Activity Discovered from Antarctica-Derived Psychrophilic Pseudogymnoascus sp. HDN17-933"

_marinedrugs, 2022, doi:10.3390/md20100593_

Round 1
Reviewer 1 Report
This article described the isolation, structural elucidation and biological activities of 6 new tetrapeptides isolated from pseudogymnoascus fungus.
Major corrections have to be done before publications of this manuscript.
First these compounds can not be named pseudopeptides A-F because pseudopeptide is a class of compounds and thiswill cause a referencing problem.
Some others points have to be change before publication :
- In introduction, activities have to be explain, for example line 42, cytotoxic activity on which cells ? IC50 ? Structural informations are added in supplementary data but there is no mention of that in the article.
- In the results, there is no mention of the HPLC chromatogram of the crude extract (S3) and in the material and methods no explanation about the analysis parameters.
- Molecular formula are not correct. For example, line 69, it must be C30H39N5O5. It is the same all the long of the manuscript.
- line 104, Figure S65 is mentioned but there is the same figure in scheme 1 that is not mention !
- line 106, there is a confusion between 1a and 1b.
- The explanation of the biological tests is really poor, there is no value (may be a table with value can be added). There is no positive control to judge the interest of these activities.
- In materials and methods, HRESIMS parameters are not described, likewise for HPLC method for crude extract analysis.
- Table 2 is not at the write place, it could be place before.
- The protocol used in part 3.8 and 3.9 have to be detailed.
Author Response
Dear Editor and Reviewers:
On behalf of my co-authors, thank you very much for giving us an opportunity to resubmit our manuscript. We appreciate editor and reviewers for the positive comments and suggestions on our manuscript entitled “Unusual tetrahydropyridoindole-containing tetrapeptides with human nicotinic acetylcholine receptors targeting activity discovered from Antarctica-derived psychrophilic Pseudogymnoascus sp. HDN17-933”. (NO.: marinedrugs-1909373). We have studied the comments carefully and have made the corresponding corrections. Revised portion and other inappropriate parts are marked in red in the paper.

Reviewer 2 Report
1. In the research method, it is necessary to add an explanation of Target/Receptor selection and preparation, Ligand selection and preparation and Evaluating docking results.
2. Data regarding binding affinity and Hydrogen bond analysis of compounds 1-6 as ligands need to be shown in the table
3. Procedures regarding the inhibition of activities of nicotinic acetylcholine receptors are incomplete and there is no reference
Author Response

(The authors gave the same response as above.)

Reviewer 3 Report
This review concerns the article type manuscript with title: Unusual tetrahydropyridoindole-containing tetrapeptides with human nicotinic acetylcholine receptors targeting activity discovered from Antarctica-derived psychrophilic Pseudogymnoascus sp. HDN17-933. The manuscript was submitted to Marine Drugs journal (Manuscript ID: marinedrugs-1909373).
My opinion is valid for structure determination. Six new compounds were isolated from psychrophilic fungus Pseudogymnoascus sp. HDN17-933 derived from Antarctica and their structures were analyzed mainly using 2D NMR spectroscopy (1H, 13C, DEPT, COSY, NOESY, HMBC, HSQC and HRESIMS) supported by Molecular Docking studies for inhibitory role of compound 2 to hα4β2. The methodology of structure determination is correct and typical for such study.
In my opinion the manuscript can be accepted for publication.
Author Response

(The authors gave the same response as above.)

Round 2
Reviewer 1 Report
The authors have taken into account the comments made previously and have improved the whole manuscript.
I think this manuscript is now acceptable for publication in this present form.